# TokenSculpt: Pruning with Min-Max Spatio-Temporal Duplication for Video Grounding

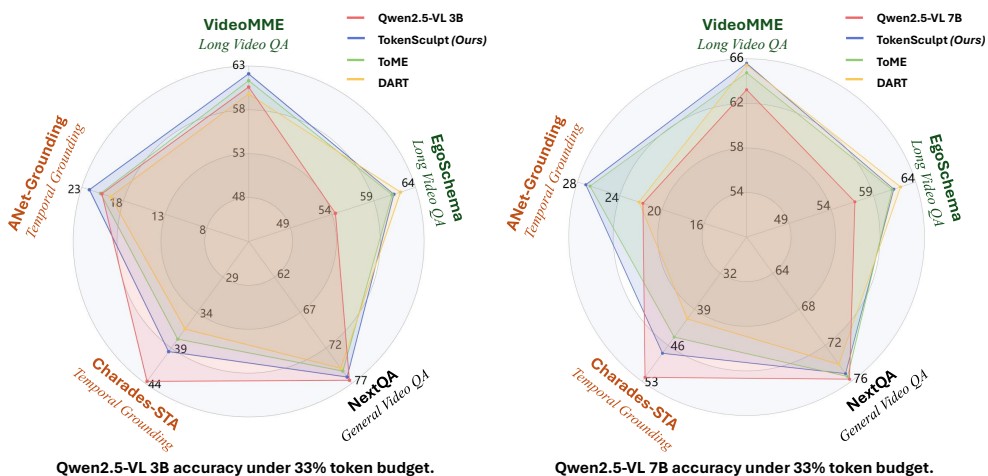

Figure 1: Comparison of training-free token reduction methods using Qwen2.5-VL 3B and Qwen2.5-VL 7B under 33% token budgets.

## Abstract

Visual token pruning is essential for reducing computational overhead in multimodal large language models (MLLMs), especially for videos where visual tokens outnumber text ones. Existing pruning methods, typically based on attention or similarity, barely consider the spatiotemporal structure of videos and may incorrectly merge low-similarity or irrelevant tokens, leading to information loss. We propose **TokenSculpt**, a structure-aware pruning approach designed for video inputs. It aggregates tokens based on similarity while explicitly avoiding low-similarity merges, and applies a bipartite matching strategy to uniformly sample tokens across spatial and temporal dimensions. This design helps preserve the structural integrity of video representations. Additionally, **TokenSculpt** is compatible with Flash-Attention, enabling efficient integration into modern MLLMs. Experiments across multiple video-language tasks show that **TokenSculpt** consistently outperforms prior methods. It achieves an average improvement of 2.9% over baselines. While particularly effective in redundant video settings, it also performs well across a range of scenarios. Our approach provides an efficient and scalable solution for video token pruning and improves performance in grounding and related video-language understanding tasks.

## 1 Introduction

Multimodal large language models (MLLMs) have achieved remarkable progress across a variety of vision tasks in real-world applications, including visual question answering (VQA), video understanding (Wang et al., 2024b), multimodal reasoning (Wang et al., 2024c), and more. However, much of this progress has been driven by a substantial increase in the number of visual tokens,

especially when handling high-resolution images or long-duration videos. This increase results in significant computational overhead, limiting inference efficiency in practice.

To address this issue, a number of recent approaches study compressing redundant visual tokens during inference for efficiency. Attention-based pruning strategies (Chen et al., 2024a; Zhang et al., 2024d; Xing et al., 2024) typically begin by computing an importance score for each visual token, and then discard the least important ones to reduce compute. These approaches rely on attention weights between visual and text tokens for relevance.

However, attention-based token pruning methods face a critical limitation: they are fundamentally incompatible with popular acceleration techniques such as FlashAttention (Dao et al., 2022). These methods typically require recomputation of the full attention to estimate token importance, incurring huge extra compute. Regarding this, recent works have proposed similarity-based token pruning strategies, including DART (Wen et al., 2025) and ToME (Bolya et al., 2022). These approaches leverage pairwise token similarity (e.g., cosine similarity) or token norm to perform token merging or discarding operations, without recomputing the full attention matrix. As a result, they offer better scalability and inference efficiency, making them more suitable for real-world deployment.

While both attention- and similarity-based pruning methods have shown promising results on general vision tasks, they often overlook spatio-temporal structure, which is critical for tasks involving strong temporal or spatial dependencies, such as video grounding.

Through detailed inspection of token merging and pruning, we observe two common failure modes:

- **Attention-based methods** tend to preserve tokens concentrated in small high-attention regions, leading to severe coverage gaps and the loss of peripheral or contextually critical areas.
- **Similarity-based methods** often suffer from the so-called *huge-cluster phenomenon*, where a number of similar tokens are merged into a single one. This effectively collapses spatially and temporally distributed information into a single point, preventing the model from reconstructing the original positional and sequential structure from the compressed representation.

These observations highlight a key challenge in token pruning: how to balance compression efficiency with the preservation of spatio-temporal structure. Addressing this challenge is essential for improving model performance in structured multimodal tasks.

To address the above limitations, we introduce **TokenSculpt**, a novel token compression framework designed to preserve spatio-temporal grounding in MLLMs. Unlike prior approaches that often discard or distort key visual structures, **TokenSculpt** explicitly models event boundaries via a mechanism called Min-Max Duplication. It identifies and retains start-end anchors along spatial and temporal dimensions. This allows the model to better localize and reconstruct regions even after aggressive token reduction. In addition, **TokenSculpt** incorporates a linearly uniform sampling strategy to ensure balanced token distribution across time and space, reducing the risk of regional collapse or token redundancy. Our method is plug-and-play, inference-efficient, and requires no additional training. Experiments on multiple vision-language benchmarks demonstrate that **TokenSculpt** achieves better trade-offs between compression rate and task performance, particularly for tasks requiring precise grounding in space and time.

Our main contributions are summarized as follows:

- We empirically demonstrate that both **linearly uniform sampling** and **start-end anchoring** are crucial for preserving essential visual information in multimodal tasks with strong spatio-temporal dependencies. These components notably enhance model alignment and representation under token compression.
- We propose the **TokenSculpt** framework and design a novel **Min-Max Duplication** mechanism, which explicitly retains start and end anchors along spatial and temporal axes. This guides token merging and effectively mitigates issues such as token misalignment, region collapse, and structural confusion. At a 33% token compression rate, our method retains **100.89%** of the original performance, outperforming existing approaches by **2.75%**.
- **TokenSculpt** is plug-and-play at inference, requiring no additional training. It consistently outperforms existing token pruning baselines across multiple video understanding and multimodal grounding benchmarks.

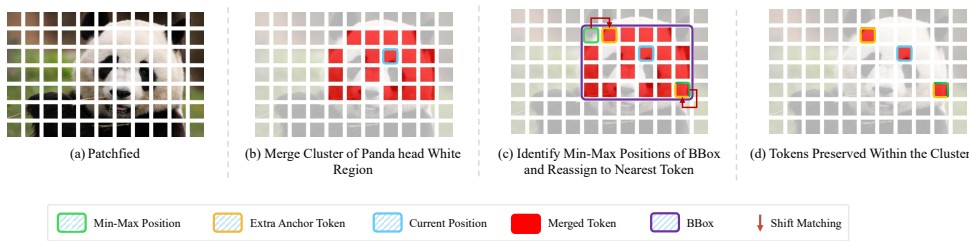

(a) Patchfied | (b) Merge Cluster of Panda head White Region | (c) Identify Min-Max Positions of BBox and Reassign to Nearest Token | (d) Tokens Preserved Within the Cluster

Min-Max Position    Extra Anchor Token    Current Position    Merged Token    BBox    Shift Matching

Figure 2: Illustration of the Min-Max Duplication framework. (a) Original image of a panda. (b) Tokens from the white region of the panda's head clustered to the blue position. (c) Bounding box around the white region with green tokens representing the start and end anchors, which are shifted to their nearest tokens (orange ones). (d) Final set of retained tokens after the Min-Max Duplication after shifting process.

## 2 RELATED WORK

**Multimodal large language models.** Recently, multimodal large language models (MLLMs) (Wang et al., 2024a; Bai et al., 2025; Chen et al., 2024b; DeepSeek-AI, 2025; Zhu et al., 2025; Li et al., 2024b) have increasingly adopted video and text as inputs to perform tasks such as grounding, understanding, and reasoning (Li et al., 2025b; Huang et al., 2024). However, with the huge increase in video resolution and duration, the computational overhead during learning and inference have become heavily concentrated in the visual modality (Li et al., 2025a; Liu et al., 2024; Yao et al., 2025; Xiong et al., 2025). Visual inputs often contain substantial redundancy, and the quadratic complexity of attention operations further amplifies this overhead (Zeng et al., 2024; Yan et al., 2024; Wang et al., 2025).

To address these issues brought by long inputs, current MLLMs primarily adopt two strategies: **(i)** extending the context window of large language models (LLMs) (as in Gemini 1.5 (Team et al., 2024), LongViLa (Chen et al., 2024c), LongVA (Zhang et al., 2024b), Qwen2.5-VL (Bai et al., 2025)); and **(ii)** applying token compression to reduce the visual processing load, as employed by models such as VideoChat-Flash (Li et al., 2024b), Qwen2.5-VL, and Q-Former (Li et al., 2023).

**Token pruning.** With the increasing demands of high-resolution and long-duration video inputs, the number of visual tokens has become significantly larger than textual tokens hundreds of times more. Moreover, visual information tends to exhibit substantial spatial and temporal redundancy. As a result, efficient compression of visual tokens is critical for reducing computational cost and enabling scalable multimodal processing.

To address this challenge, several methods have been proposed. For instance, VideoChat2 (Li et al., 2024a) and InternVL(Zhu et al., 2025) leverage Q-Former with context to compress visual tokens, while DeCO (Yao et al., 2024) adopts adaptive pooling to dynamically reduce tokens. However, these approaches generally require additional training or fine-tuning, limiting their generalization and plug-and-play deployment across different models (Song et al., 2024; Zhang et al., 2024a; Shu et al., 2025). Other approaches attempt to perform token pruning without extra training. ToME (Bolya et al., 2022) and DART (Wen et al., 2025) merges tokens based on similarity metrics, while FastV (Chen et al., 2024a), SparseVLM (Zhang et al., 2024d) and PyramidDrop (Xing et al., 2024) select informative tokens using attention scores. However, the attention-based methods are typically incompatible with efficient attention operators like FlashAttention (Dao et al., 2022), which restricts their efficiency in modern transformer architectures.

## 3 METHODOLOGY

In this section, we illustrate the proposed **TokenSculpt**, an efficient token compression scheme to help MLLMs address visual grounding tasks. Specifically, **TokenSculpt** incorporates a novel segmentation and localization operation. This approach not only addresses the token ordering issue but also enhances spatiotemporal modeling. The additional segmentation and localization operations

introduce only a minimal amount of negligible extra computation, yet they significantly enhance the model's performance on grounding tasks after compression.

## 3.1 PRELIMINARIES

**Architecture of MLLM.** Multimodal Large Language Models (MLLM) typically consist of three components: Visual Encoder, Modality Projector, and Large Language Model (LLM). For a given visual-text input, the visual encoder first splits the image or video into patches and processes them to obtain the corresponding visual embeddings. These visual embeddings are then mapped to the semantic space of the LLM via the modality projector, resulting in the visual token $E_v$. The visual token $E_v$ is concatenated with the text token $E_t$, which has been processed by the LLM. The model then uses the semantically aligned tokens $E_v$ and $E_t$ as input to generate the output.

**Computational complexity of MLLMs.** Recent advancements in MLLMs asks for the increase in visual token number, as in works like LLaVA (Liu et al., 2023). As a result, visual tokens have become the dominant factor that contributes to the computational cost of MLLMs. The total FLOPs (floating-point operations) for MLLMs can be expressed as:

$$\text{Total FLOPs} = T \times (4nd^2 + 2n^2d + 2ndm),$$

where $T$ denotes the number of transformer layers, $n$ is the sequence length (primarily determined by visual tokens), $d$ is the hidden size of the LLM, and $m$ is the intermediate size of the Feedforward Neural Network (FNN). This equation shows that when $n$ is large, the visual tokens significantly impact the overall computational complexity.

## 3.2 TOKEN SCULPT FOR POSITION-PRESERVING PRUNING

To enable MLLMs to reserve spatiotemporal modeling even after explicitly compressing its fed visual tokens, we propose **TokenSculpt**, a three-step position-friendly token pruning approach. It firstly finds a few anchor tokens with crucial position information using min-max duplication (a self-proposed operation). Then it roughly samples the given tokens uniformly, and conducts a min-max guided similarity-based merging, reducing tokens into a few. Finally, it sorts these merged tokens based on their original locations.

**Anchor selection via min-max duplication.** In visual grounding related tasks, precise understanding of some activities and motivation often relies on accurate localization of an event, action, and object, involving when and where they begin and end. Motivated by this observation, we propose a operation called Min-Max Duplication to preserve such critical position information during token merging.

For humans, recognizing and localizing an action typically involves understanding its temporal span (when it starts and ends) and its spatial extent (where it happens in the scene). These two aspects can be compactly represented by the minimum and maximum values in both temporal and spatial dimensions. Instead of retaining all tokens uniformly, we selectively duplicate those that encode this boundary information, ensuring that the merged representation retains a strong grounding signal.

As illstrated in Figure. 2, given a set of token coordinates $\mathbf{p}_i = (t_i, x_i, y_i)$, where $t_i$ denotes the temporal position and $(x_i, y_i)$ the spatial location, we define the min-max coordinate pair as:

$$\mathbf{p}_i = (t_i, x_i, y_i), \quad \mathbf{p}_{\min} = \min_i \mathbf{p}_i, \quad \mathbf{p}_{\max} = \max_i \mathbf{p}_i.$$

where $\mathbf{p}_{\min}$ is the earliest and top-left event boundary, while $\mathbf{p}_{\max}$ is the latest and bottom-right one.

To retain fine-grained localization information, we find the original token whose position is closest to each boundary and duplicate it as an anchor token. These anchors serve as hard constraints in the token merging process, ensuring that no matter how aggressively we merge other tokens, the min-max anchors are always retained.

We define the anchor tokens as:

$$\mathbf{p}_{\min \text{ anchor}} = \arg\min_i \|\mathbf{p}_i - \mathbf{p}_{\min}\|_2, \quad \mathbf{p}_{\max \text{ anchor}} = \arg\min_i \|\mathbf{p}_i - \mathbf{p}_{\max}\|_2.$$

**Token merging.** In vision-language models, visual encoders typically produce a large number of dense tokens from input images or videos. Directly processing all of these tokens not only introduces considerable computational overhead but also risks retaining redundant or uninformative content, especially in long videos or high-resolution scenes. To address this, we propose a two-stage similarity-based token merging framework **TokenSculpt**, designed to effectively reduce token redundancy while preserving critical spatio-temporal information.

- Stage 1: Uniform Sampling for Initial Compression. We begin by applying uniform sampling to the full set of visual tokens. This step operates solely based on token positions and ignores semantics. By retaining tokens at regular spatial and/or temporal intervals, we achieve a lightweight compression that reduces the token count while maintaining global structure. Although this process lacks fine-grained adaptivity, it provides a fast approximation and lays the foundation for the more selective merging that follows.

- Stage 2: Min-Max Guided Similarity-Based Merging. After the initial compression, we introduce Min-Max Duplication to reinsert anchor tokens corresponding to key spatio-temporal boundaries. These anchors ensure that vital positional information is preserved. With this enriched token set, we proceed to the second stage: a similarity-based Token Merging step inspired by ToMe.

In Stage 2, we get pairwise cosine similarity between each non-anchor source token and rest as:

$$\mathbf{Sim}_{ij} = \frac{\mathbf{s}_i^\top \mathbf{a}_j}{\|\mathbf{s}_i\|_2 \|\mathbf{a}_j\|_2}.$$

where $\mathbf{s}_i$ is a source token and $\mathbf{a}_j$ is an anchor token. Each source token is assigned to its most similar anchor token, forming a point cluster. Within each cluster, we perform average pooling to merge tokens and generate new, representative token embeddings:

$$\hat{\mathbf{t}}_k = \frac{1}{|\mathcal{C}_k|} \sum_{i \in \mathcal{C}_k} \mathbf{t}_i,$$

where $\mathcal{C}_k$ denotes the cluster associated with anchor $k$, and $\hat{\mathbf{t}}_k$ is the resulting merged token.

**Post-merging reordering.** To maintain the integrity of spatio-temporal alignment, we sort the merged tokens based on their original positions after merging. This step ensures that the final token sequence maintains a coherent temporal and spatial structure, which is especially critical for downstream grounding tasks that rely on precise ordering.

**Analysis.** Compared with most existing attention-based token pruning means, our **TokenSculpt** is built upon similarity-based ones, considering the former ones are less practical in applications. Since attention-based methods often introduce non-trivial computational overhead due to the need for pairwise attention computation. Moreover, it is difficult to integrate them with modern acceleration frameworks (e.g. Flash-Attention), which are optimized for fixed-sized attention patterns.

Regarding our **TokenSculpt** is similarity-based, it is inherently more efficient and hardware-friendly. Specifically, it leverages both coarse-level uniform sampling and a fine-grained min-max duplication mechanism to preserve critical spatiotemporal information in the merging process.

### 3.3 TOKENSCULPT IN PRACTICE.

To understand how **TokenSculpt** performs under real-world constraints, we analyze its behavior on existing benchmarks, and discuss why current solutions barely address visual grounding problems. On the other hand, most current benchmarks fall short in evaluating spatial and temporal alignment. Existing token pruning methods are primarily evaluated on general visual benchmarks such as GQA (Hudson & Manning, 2019), VQAv2 (Tang et al., 2019), VQAText (Wang & Ji, 2018), OCRBench, and MVBench (Li et al., 2024a). These benchmarks tend to focus on recognizing what is present in the visual content, while neglecting spatial relationships or temporal dynamics. In the following, we analyze the disconnect between current benchmarks and existing token pruning approaches in details.

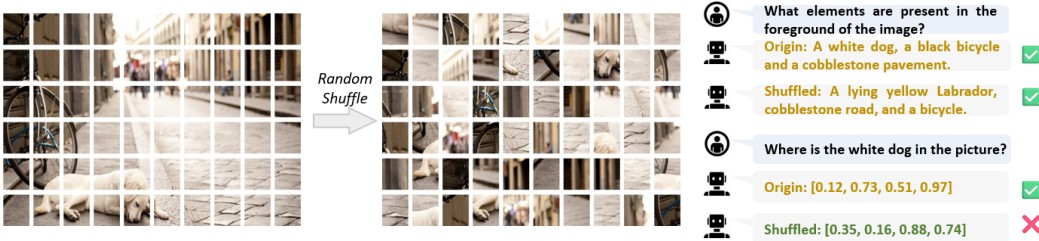

Figure 3: An illustration of token order sensitivity in MLLMs. The original image (left) and its randomly shuffled counterpart (right) are used as inputs to MLLMs. While both versions yield semantically correct foreground descriptions, the shuffled input degrades localization performance.

**General Multimodal Benchmarks Barely Measure "Where" or "When".** General visual benchmarks play a crucial role in evaluating MultiModal Large Language Models (MLLMs), offering systematic assessments of their capabilities in understanding, perception, and reasoning. However, the majority of these benchmarks tend to focus on coarse-grained recognition tasks, such as object identification in an image, while placing much less emphasis on fine-grained comprehension, including when events occur or where objects are located.

To investigate the model's sensitivity to token order, we experiment where we randomly shuffled all tokens in MVBench (Li et al., 2024a) inputs before feeding them into the MLLMs. Surprisingly, the model performance drops by only about 3%, indicating that the token order has limited impact on many tasks in MVBench (Li et al., 2024a).

In stark contrast, when we apply the same token-shuffling strategy to grounding task, the model performance decreased by more 60%. This reveals a critical difference: fine-grained grounding tasks rely heavily on the sequential structure of tokens, unlike coarse-level understanding tasks. We could oberserve similar phenomenon in Figure. 3, even after shuffling all visual tokens in an image, the model remains capable of answering perception and understanding questions. Yet, it completely fails when tackling grounding questions.

These results highlight a significant gap in current visual benchmarks: they often assess "what is seen" but overlook "where" and "when" information. This underscores the need for benchmarks that more rigorously evaluate spatio-temporal alignment and fine-grained grounding capabilities.

**Why Most Token Pruning Methods Struggles with "Where" or "When".** Based on our findings, we analyze the limitations of current leading token pruning methods. We suppose they mainly stem from attention-based bottom-heavy bias and similarity-based over-aggregation.

*Attention-based bottom-heavy bias.* Attention-based methods rely on computing attention scores and selecting the top-k tokens with the highest values. The attention score is defined as:

$$\text{Attention}(Q, K, V) = \text{softmax}\left(\frac{\mathbf{Q} \cdot \mathbf{K}^\top}{\sqrt{d_k}}\right) \cdot \mathbf{V},$$

$$\text{Attn Score}(x_i) = \frac{1}{N}\sum_{j=1}^{N}\text{Attention}(x_i, x_j).$$

However, in pruning methods based on attention scores, we find that their effectiveness is severely constrained by network depth. In the shallow layers of the network, the attention weights have not yet stabilized or converged, making it difficult for the model to accurately identify and focus on critical tokens. This directly leads to suboptimal pruning decisions in the early stages.

More critically, we reveal that such methods are significantly influenced by position embeddings (Endo et al., 2024; Zhang et al., 2024c), exhibiting a systematic and strong preference for preserving information from the bottom of the image. As shown in Figure. 4, this inherent "bottom-heavy bias" not only undermines the method's generalizability and reliability, but also causes a substantial degradation in performance on tasks that are highly sensitive to spatial information, such as visual grounding.

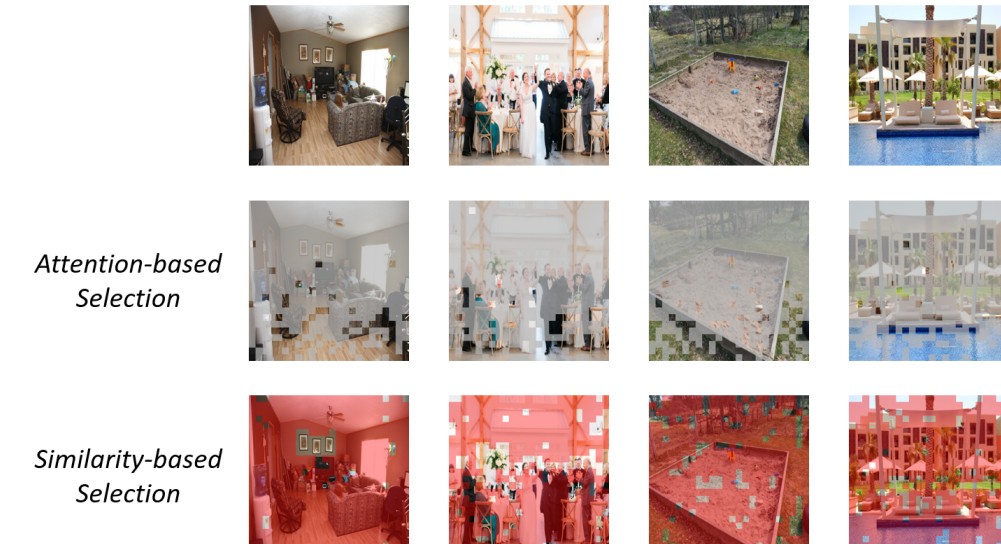

Figure 4: Illustration of spatial biases in token pruning methods. Attention-based pruning exhibits a strong bottom-heavy bias, preferentially retaining tokens from the lower image regions due to unstable early-layer attention and positional embedding interference. Similarity-based pruning shows over-aggregation, where most tokens rapidly merge into a few dominant clusters, leading to structural collapse and loss of fine-grained spatial details.

*Similarity-based over-aggregation.* For similarity-based pruning methods (e.g. ToME (Bolya et al., 2022)), we conduct a key ablation study. By directly merging tokens iteratively based on a global similarity matrix, our experiments reveal a rapid and unbalanced aggregation phenomenon. The vast majority of tokens are quickly merged into a few super-clusters, while the remaining tokens are relegated to several small discrete units.

As illustrated in Figure. 4, this highly centralized aggregation pattern indicates that relying solely on similarity scores leads to the over-compression of similar tokens into a few dominant clusters, resulting in a catastrophic loss of critical spatio-temporal structures in those regions.

## 4 EXPERIMENTS.

We evaluate our **TokenSculpt** on a series of MLLMs (including Qwen2.5-VL) and VideoChat-R1 (Li et al., 2025b) across three image grounding benchmarks and four video grounding benchmarks. All experiments are conducted with eight NVIDIA A100 GPUs. Please refer to the Appendix for implementation details.

### 4.1 MAIN RESULTS.

**Video Tasks.** To validate the effectiveness of **TokenSculpt** in video token pruning, we evaluate our approach on five benchmarks: Charades-STA (Gao et al., 2017), ANet-Grounding (Krishna et al., 2017), VideoMME (Fu et al., 2025), EgoSchema (Mangalam et al., 2023), and Next-QA (Xiao et al., 2021). For video grounding tasks, the results shown in Table 1 demonstrate that **Token-Sculpt** achieves significant performance improvements. Under low token budget, **TokenSculpt** retains 90.04%, 102.49%, and 106.61% of the performance on Charades-STA, VideoMME, and EgoSchema, respectively. We further make the following observations:**(i)** Our method achieves an average improvement of 3.98% across the video grouding benchmarks, significantly enhancing the model's grounding capability. **(ii)** As the token compression ratio increases, our method exhibits more robust performance than ToME at lower retention rates. **(iii)** Our method shows strong generalizability and can be seamlessly extended to larger-scale multimodal models. Experiments on larger models such as Qwen2.5-VL-Max demonstrate that our method can be directly applied without structural modifications, maintaining excellent performance in both compression ratio and grounding accuracy, indicating promising scalability and practical applicability.

| Method | Charades-STA | | | | ANet-Grounding | | | | VideoMME | | | | EgoSchema | Next-QA | Overall |
|---|---|---|---|---|---|---|---|---|---|---|---|---|---|---|---|
| | R@0.3 | R@0.5 | R@0.7 | mIoU | R@0.3 | R@0.5 | R@0.7 | mIoU | short | medium | long | Avg | | | |
| **Qwen2.5-VL-3B** | | | | | | | | | Retain 100% Tokens | | | | | | |
| baseline | 66.57 | 46.26 | 22.47 | 43.70 | 28.56 | 15.94 | 7.55 | 20.59 | 73.20 | 58.00 | 50.70 | 60.60 | 54.40 | 76.55 | 100.00% |
| **Qwen2.5-VL-3B** | | | | | | | | | Retain 66% Tokens | | | | | | |
| FastV | 60.65 | 40.46 | 18.79 | 39.55 | | OOM | | | | OOM | | | OOM | OOM | - |
| DART | 63.93 | 44.49 | 21.37 | 42.14 | 28.75 | 16.06 | 7.59 | 20.84 | 71.60 | 58.60 | 50.10 | 60.07 | 56.00 | 76.24 | 99.71% |
| ToME | 63.60 | 43.36 | 20.81 | 41.84 | 28.48 | 15.77 | 7.51 | 20.58 | 72.90 | 57.80 | 50.90 | 60.52 | 57.00 | 76.33 | 99.74% |
| **TokenSculpt** | 64.65 | 44.73 | 20.83 | 42.31 | 29.71 | 16.58 | 8.01 | 21.33 | 73.20 | 58.60 | 51.20 | 61.00 | 57.20 | 76.44 | 101.23% |
| **Qwen2.5-VL-3B** | | | | | | | | | Retain 49% Tokens | | | | | | |
| FastV | 56.56 | 37.85 | 17.18 | 37.05 | | OOM | | | | OOM | | | OOM | OOM | - |
| DART | 60.70 | 42.04 | 19.60 | 39.98 | 27.90 | 15.38 | 7.39 | 20.42 | 70.80 | 58.30 | 51.20 | 60.11 | 57.40 | 75.01 | 98.50% |
| ToME | 60.81 | 41.48 | 19.95 | 40.39 | 29.51 | 16.65 | 7.57 | 21.20 | 73.40 | 58.20 | 51.40 | 61.04 | 57.80 | 75.28 | 100.03% |
| **TokenSculpt** | 63.68 | 43.55 | 20.27 | 41.63 | 29.98 | 16.72 | 7.92 | 21.61 | 73.30 | 59.70 | 51.90 | 61.63 | 58.40 | 75.72 | 101.85% |
| **Qwen2.5-VL-3B** | | | | | | | | | Retain 33% Tokens | | | | | | |
| FastV | 51.10 | 31.86 | 14.52 | 32.98 | | OOM | | | | OOM | | | OOM | OOM | - |
| DART | 55.05 | 36.72 | 16.86 | 36.33 | 26.72 | 14.50 | 6.41 | 19.56 | 67.90 | 60.00 | 51.60 | 59.81 | 58.80 | 74.85 | 95.61% |
| ToME | 57.10 | 37.55 | 17.77 | 37.75 | 28.84 | 15.91 | 7.07 | 20.75 | 72.20 | 60.00 | 51.80 | 61.33 | 58.40 | 75.26 | 98.13% |
| **TokenSculpt** | 59.65 | 39.95 | 19.36 | 39.52 | 30.66 | 17.35 | 7.90 | 22.09 | 72.40 | 61.00 | 52.90 | 62.11 | 58.00 | 76.10 | 100.89% |
| **Qwen2.5-VL-7B** | | | | | | | | | Retain 100% Tokens | | | | | | |
| baseline | 76.02 | 60.24 | 33.74 | 52.21 | 28.77 | 16.96 | 8.82 | 21.81 | 75.70 | 61.00 | 53.00 | 63.22 | 56.80 | 75.75 | 100.00% |
| **Qwen2.5-VL-7B** | | | | | | | | | Retain 66% Tokens | | | | | | |
| FastV | | OOM | | | | OOM | | | | OOM | | | OOM | OOM | - |
| DART | 73.36 | 55.16 | 30.38 | 49.50 | 31.85 | 18.45 | 9.85 | 24.15 | 76.00 | 63.00 | 53.60 | 64.19 | 60.60 | 75.55 | 102.43% |
| ToME | 74.68 | 57.50 | 30.94 | 50.52 | 32.14 | 18.73 | 9.85 | 24.12 | 75.90 | 62.40 | 54.30 | 64.22 | 58.80 | 75.72 | 102.46% |
| **TokenSculpt** | 75.11 | 58.66 | 32.10 | 51.14 | 32.80 | 19.38 | 10.29 | 24.80 | 76.10 | 64.10 | 54.20 | 64.81 | 59.20 | 75.85 | 103.77% |
| **Qwen2.5-VL-7B** | | | | | | | | | Retain 49% Tokens | | | | | | |
| FastV | | OOM | | | | OOM | | | | OOM | | | OOM | OOM | - |
| DART | 68.15 | 50.27 | 25.81 | 45.82 | 31.45 | 17.90 | 9.32 | 23.87 | 75.10 | 64.70 | 54.60 | 64.78 | 61.20 | 75.01 | 100.57% |
| ToME | 72.63 | 54.52 | 28.95 | 48.83 | 34.11 | 20.04 | 10.59 | 25.61 | 75.70 | 63.80 | 54.40 | 64.63 | 60.00 | 75.28 | 103.56% |
| **TokenSculpt** | 72.93 | 55.89 | 29.87 | 49.57 | 34.78 | 20.60 | 11.04 | 26.06 | 76.90 | 63.90 | 53.90 | 64.89 | 59.60 | 75.72 | 104.47% |
| **Qwen2.5-VL-7B** | | | | | | | | | Retain 33% Tokens | | | | | | |
| FastV | | OOM | | | | OOM | | | | OOM | | | OOM | OOM | - |
| DART | 61.34 | 43.20 | 21.29 | 40.87 | 28.84 | 16.43 | 8.11 | 22.18 | 75.60 | 65.30 | 55.40 | 65.44 | 62.20 | 74.11 | 96.97% |
| ToME | 67.28 | 46.94 | 23.31 | 44.40 | 36.12 | 21.45 | 11.44 | 26.81 | 74.70 | 64.90 | 54.70 | 64.74 | 61.20 | 75.46 | 103.43% |
| **TokenSculpt** | 70.70 | 52.85 | 27.12 | 47.54 | 36.44 | 21.79 | 11.64 | 27.21 | 75.00 | 66.00 | 55.80 | 65.59 | 61.40 | 75.14 | 105.48% |
| **VideoChat-R1 7B** | | | | | | | | | Retain 100% Tokens | | | | | | |
| baseline | 81.64 | 69.35 | 44.44 | 58.37 | 40.03 | 25.16 | 13.33 | 28.13 | 70.20 | 56.60 | 48.90 | 58.56 | 55.60 | 76.59 | 100.00% |
| **VideoChat-R1 7B** | | | | | | | | | Retain 66% Tokens | | | | | | |
| FastV | | OOM | | | | OOM | | | | OOM | | | OOM | OOM | - |
| DART | 69.70 | 51.88 | 29.76 | 47.30 | 50.17 | 31.67 | 16.43 | 34.28 | 74.90 | 61.00 | 52.00 | 62.62 | 62.40 | 75.60 | 103.61% |
| ToME | 79.92 | 63.87 | 37.02 | 55.07 | 59.27 | 39.22 | 22.45 | 40.93 | 75.70 | 63.70 | 54.40 | 64.59 | 60.40 | 75.74 | 112.05% |
| **TokenSculpt** | 79.41 | 65.22 | 38.49 | 55.75 | 59.06 | 39.91 | 22.61 | 41.16 | 74.90 | 64.30 | 54.00 | 64.41 | 63.00 | 75.89 | 113.34% |
| **VideoChat-R1 7B** | | | | | | | | | Retain 49% Tokens | | | | | | |
| FastV | | OOM | | | | OOM | | | | OOM | | | OOM | OOM | - |
| DART | 77.28 | 62.66 | 37.15 | 53.88 | 51.92 | 34.11 | 18.10 | 35.89 | 74.90 | 63.30 | 53.90 | 64.04 | 60.60 | 76.05 | 107.53% |
| ToME | 80.86 | 66.77 | 39.76 | 56.77 | 53.77 | 35.39 | 19.55 | 37.33 | 75.40 | 63.70 | 53.30 | 64.15 | 60.60 | 76.84 | 109.94% |
| **TokenSculpt** | 80.94 | 68.23 | 41.96 | 57.37 | 53.20 | 35.79 | 19.66 | 37.23 | 75.30 | 62.80 | 53.70 | 63.93 | 60.40 | 76.21 | 109.94% |
| **VideoChat-R1 7B** | | | | | | | | | Retain 33% Tokens | | | | | | |
| FastV | | OOM | | | | OOM | | | | OOM | | | OOM | OOM | - |
| DART | 80.05 | 67.45 | 42.58 | 57.02 | 48.52 | 31.75 | 17.02 | 33.80 | 75.70 | 63.00 | 53.80 | 64.15 | 61.20 | 76.62 | 107.52% |
| ToME | 81.45 | 68.82 | 43.52 | 58.01 | 47.57 | 30.86 | 16.92 | 33.15 | 75.30 | 61.80 | 52.10 | 64.56 | 59.00 | 76.49 | 106.74% |
| **TokenSculpt** | 81.53 | 68.84 | 43.28 | 58.08 | 48.98 | 32.36 | 17.53 | 34.24 | 75.20 | 62.70 | 53.20 | 63.70 | 58.80 | 76.70 | 107.29% |

Table 1: Comparative experiments on video grounding related tasks. In all experiments for **Token-Sculpt**, tokens are pruned after the second layer.

| Method | Token (%) | Total Time (Min:Sec) | Prefill Time (Min:Sec) | TFLOPs | Cuda Memory (MB) | Speedup | |
|---|---|---|---|---|---|---|---|
| | | | | | | (Total) | (Prefill) |
| Qwen2.5-VL-7B | 100% | 67:03 | 66:04 | 930.65 | 34791 | 1.00x | 1.00x |
| DART | 33% | 37:00 | 36:30 | 506.93 | 24936 | 1.81x | 1.81x |
| ToME | 33% | 35:11 | 34:39 | 506.96 | 24274 | 1.90x | 1.90x |
| **TokenSculpt** | 33% | 35:43 | 35:18 | 506.93 | 24274 | 1.88x | 1.87x |

Table 2: Efficiency analysis of **TokenSculpt** on Qwen2.5-VL 7B with 33% tokens retained.

## 4.2 ABLATION STUDY AND ANALYSIS.

**Efficiency Analysis.** As shown in Table 2, we compare the prefill time of ToME (Bolya et al., 2022), DART(Wen et al., 2025) and **TokenSculpt**. The prefill time of **TokenSculpt** is 84% of origin and 74% of DART. More detailed results are provided in E.

**Influence from Sampling Strategies.** In **TokenSculpt**, we adopt a linearly uniform sampling strategy to select representative visual tokens, aiming to preserve the spatio-temporal structure of videos during the token compression process. Compared to approaches that rely on model attention weights or heuristic token importance scoring, uniform sampling exhibits stronger generalization and incurs significantly lower computational overhead.

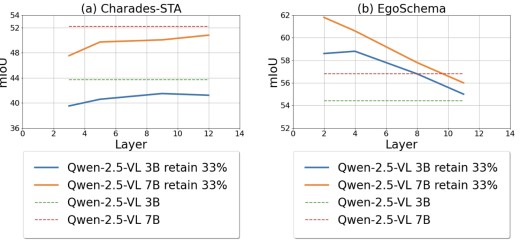

Figure 5: Effect of layer on token pruning.

| Method | Charades-STA | | | |
|---|---|---|---|---|
| | R@0.3 | R@0.5 | R@0.7 | mIoU |
| uniform | 59.65 | 39.95 | 19.36 | 39.52 |
| random | 58.60 | 38.60 | 17.66 | 38.45 |
| pivot | 50.32 | 33.11 | 16.07 | 33.66 |

Table 3: Sample strategy analysis of **TokenSculpt** on Qwen2.5-VL 3B with 33% of tokens retained.

To evaluate the impact of different sampling strategies on model performance, we compare three schemes in Table 3: random sampling, importance-based sampling, and our proposed uniform sampling(see Appendix E for more detailed results). Experimental results show that uniform sampling achieves the best performance without introducing any additional computational cost. This suggests that: **(i)** sophisticated token importance estimation strategies are not necessarily effective in the context of highly redundant visual representations; and **(ii)** uniform sampling provides more balanced coverage across temporal and spatial dimensions, effectively reducing information loss, especially in long-form video scenarios.

**Influence from Min-Max Duplication.** As shown in Table 1, removing the Min-Max Duplication module from **TokenSculpt** effectively reduces the method to a ToME-like baseline. We observe a performance drop ranging from 1% to 4% across different tasks when Min-Max Duplication is removed. This highlights the importance of the duplication mechanism.

The Min-Max Duplication strategy is inspired by human perceptual patterns: by identifying both the starting and ending anchors of relevant content, it guides the model to better localize and retain critical spatio-temporal regions. This design proves particularly beneficial for tasks with strong temporal or spatial dependencies, where precise alignment and structural awareness are crucial for accurate grounding.

**Influence from Pruned Layer.** We systematically investigate the impact of pruning layer selection on model performance using the Charades-STA and EgoSchema benchmarks, as shown in Figure. 5. The experimental results reveal a clear trend: as pruning is applied progressively from shallow to deeper layers, model performance initially improves and quickly saturates. Specifically, when pruning is conducted around the 5th layer, the model already recovers most of its original performance. Pruning beyond this depth yields diminishing returns, indicating that mid-to-shallow layers contain most of the redundant tokens and are thus the most effective regions for pruning.

Interestingly, on the long-video dataset EgoSchema, we observe a distinct pattern where deeper pruning layers actually lead to performance degradation. This suggests that pruning at shallower layers plays a more crucial role in removing highly redundant or noisy information from lengthy visual inputs. Retaining low-quality tokens in early layers may propagate inefficiencies in later processing, hindering the model's ability to perform cross-modal grounding. Therefore, careful selection of the pruning layer is essential—not only for maximizing pruning efficiency, but also for maintaining or even improving performance in complex multimodal temporal tasks.

**Limitation.** Our Min-Max Duplication introduces a linear merging overhead, which may incur an additional delay of about 0.6s in extremely long cases (e.g., 10,000 frames).

## 5 CONCLUSION

We present **TokenSculpt**, a novel framework for structure-aware token compression in multimodal language models. Unlike previous approaches that rely solely on similarity-based heuristics, **TokenSculpt** introduces a Min-Max Duplication mechanism that explicitly preserves critical anchors along spatial and temporal dimensions. This enables more stable and semantically aligned token merging, effectively addressing issues such as region collapse and structural confusion. Extensive experiments demonstrate that **TokenSculpt** achieves superior performance under aggressive compression ratios, maintaining high accuracy across grounding-intensive benchmarks while significantly reducing token redundancy. Our results suggest that preserving structure, rather than merely measuring similarity, is crucial for robust multimodal understanding with compression.

ETHICS STATEMENT

All authors have thoroughly reviewed and confirmed adherence to the ICLR Code of Ethics during both the research process and the paper submission. The datasets used in this work are either publicly available benchmark resources or synthetically generated solely for research purposes. The proposed methods and findings are aimed at advancing scientific understanding of machine learning models, without involving direct risks of misuse or potential harm. We are committed to the principles of fairness, transparency, and accountability, and we encourage the responsible use of the datasets and techniques introduced in this study.

REPRODUCIBILITY STATEMENT

We have taken comprehensive measures to ensure the reproducibility of our benchmarks and experimental results. Appendices B and C provide a detailed description of the experimental setup, including all hyperparameter choices and implementation details necessary to replicate the results. To further facilitate reproducibility, we plan to release all code, model weights, and datasets under an appropriate open-source license in the future.

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

## A    THE USAGE OF LARGE LANGUAGE MODELS (LLMs)

Large Language Models (LLMs) were employed to assist with proofreading and language polishing of the manuscript. All material has been carefully verified to preserve the authors' original intent and to avoid any factual errors or unintended hallucinations from the models.

## B    DETAILS OF EXPERIMENTAL SETUP

All experiments are conducted using eight NVIDIA A100 GPUs with Python 3.10, based on the `lmms-eval` framework. The size bound in our experiments is set to 8, and the `split_num` is constrained to no more than 20% of the current number of anchors.

## C    IMPLEMENTATION DETAILS

### C.1    IMPLEMENTATION OF MIN-MAX DUPLICATION

---

**Algorithm 1** PyTorch snippet of Min-Max Duplication.

---

```
unm_idx, src_idx, dst_idx = merge_idx_info
unm_idx = unm_idx.reshape(-1)
src_idx = src_idx.reshape(-1)
dst_idx = dst_idx.reshape(-1)
source = spatial_idx_record[~dst_mask]
destination = [[i] for i in spatial_idx_record[dst_mask].tolist()]
for idx_idx in range(src_idx.shape[0]):
    destination[dst_idx[idx_idx]].append(source[src_idx[idx_idx]])
size_info = torch.tensor([len(sublist) for sublist in destination]).long().to(device)
size_info_mask = (size_info >= size_bound)
size_indices = torch.topk(size_info, split_num).indices
size_indices, _ = size_indices.sort()
extra_idx = []
for size_split_idx in range(size_indices.shape[0]):
    size_idx = size_indices[size_split_idx]
    idx_data = torch.tensor(destination[size_idx]).long().to(device)
    pos_ids = position_ids_tomerge[0][idx_data]
    pos_ids_min = pos_ids.min(dim=-2).values.reshape(-1)
    pos_ids_max = pos_ids.max(dim=-2).values.reshape(-1)
    dist_to_min = torch.norm((pos_ids - pos_ids_min).float(), dim=1)
    dist_to_max = torch.norm((pos_ids - pos_ids_max).float(), dim=1)
    extra_idx.append(idx_data[dist_to_min.argmin()])
    extra_idx.append(idx_data[dist_to_max.argmin()])
extra_idx = torch.tensor(extra_idx).long().to(device)
dst_mask[extra_idx] = True
```

---

The *Min-Max Duplication* algorithm 1 enhances spatial token preservation by duplicating boundary-representative tokens from oversized clusters. It first aggregates tokens based on merging indices, then selects clusters exceeding a predefined size threshold. For each such cluster, it identifies tokens with start and end positions, and duplicates the ones closest to these extrema. This strategy retains spatial extremes and mitigates information loss during aggressive token merging, thereby improving the robustness of downstream representations.

### C.2    IMPLEMENTATION OF TOKEN MERGE

The *bipartite soft matching* algorithm 2 performs a soft, similarity-based token merging strategy inspired by bipartite matching. Given a normalized feature metric, it computes pairwise similarity scores between source and destination tokens using dot products. Tokens to be merged are selected based on top similarity scores, while the remaining ones are preserved as unmerged. The associated merge function supports mean-based aggregation of source tokens into their matched destinations using scatter_reduce, and returns the merged sequence alongside unmerged tokens. This approach enables efficient and differentiable token reduction while preserving semantic alignment between source and destination representations.

## D    DATASETS

**Charades-STA**    Charades-STA is a temporally annotated extension of the Charades dataset, constructed for the task of temporal activity localization via language queries. It consists of over 10000

**Algorithm 2** PyTorch bipartite soft matching

```python
def bipartite_soft_matching(metric, r, dst_mask):
    """
    metric size is [batch, tokens, channels].
    r indicates the number of tokens to remove
    """
    prevent_num = 0
    with torch.no_grad() :
        metric = metric / metric.norm(dim=-1 , keepdim=True)
        a , b = metric[..., ~dst_mask, :] , metric[..., dst_mask, :]
        scores = a @ b.transpose(-1 , -2)
        node_max , node_idx = scores.max(dim=-1)
        edge_idx = node_max.argsort(dim=-1, descending=True)[..., None]
        unm_idx = edge_idx[..., r:, :]      # Unmerged Tokens
        src_idx = edge_idx[..., :r, :]      # Merged Tokens
        dst_idx = node_idx[..., None].gather(dim=-2 , index=src_idx)
    def merge(x, mode="mean", dst_select_only=False):
        src, dst = x[..., ~dst_mask, :] , x[..., dst_mask, :]
        n, t1, c = src.shape
        unm = src.gather(dim=-2 , index=unm_idx.expand(n, t1-r, c))
        if not dst_select_only :
            src = src.gather(dim=-2, index = src_idx.expand(n, r, c))
            dst = dst.scatter_reduce(-2, dst_idx.expand(n, r, c), src, reduce=mode)
        return torch.cat([unm, dst], dim = 1)
    return merge, (unm_idx, src_idx, dst_idx)
```

videos spanning 157 action categories. In some videos, structured and complex queries are used, where each query contains at least two clauses. The corresponding temporal segments for these queries typically cover less than half the total video duration, increasing the difficulty of the task.

The dataset includes 13898 training samples and 4233 testing samples, among which 1378 samples are complex queries involving multiple clauses.

**ANet-Grounding** The ANet-Grounding dataset is widely used for temporal sentence grounding tasks. It consists of approximately 20000 videos sourced from the ActivityNet dataset, each annotated with multiple temporally localized natural language descriptions. On average, each video—up to 10 minutes in length—is paired with 3.65 descriptions that refer to distinct temporal segments. These segments may overlap due to concurrent events, highlighting the dataset's capacity to model long-term and complex temporal dependencies.

**VideoMME** Video-MME is a comprehensive benchmark designed to evaluate the video understanding capabilities of multi-modal large language models (MLLMs). It includes 900 videos totaling 254 hours and 2700 human-annotated question-answer pairs. The benchmark spans a wide range of visual domains, temporal durations, and data modalities.

**EgoSchema** EgoSchema is a long-form video question-answering dataset and benchmark designed to evaluate the long-range temporal understanding abilities of vision-language models. It is constructed from the Ego4D dataset and contains over 5000 human-curated multiple-choice question-answer pairs, covering more than 250 hours of egocentric video data. The content spans a wide range of natural human activities and behaviors. Each question is associated with a 3-minute video clip and requires selecting the correct answer from five given options.

**Next-QA** NExT-QA consists of a total of 5440 videos, each with an average duration of 44 seconds. The dataset includes approximately 52000 manually annotated question-answer pairs, categorized into three types: causal (48%), temporal (29%), and descriptive (23%) questions. Each video is annotated with around 10 questions, covering a diverse range of content to facilitate comprehensive evaluation of video reasoning capabilities.

## E EFFICIENCY ANALYSIS.

To comprehensively evaluate the computational and memory efficiency of various token reduction methods, we report detailed metrics across multiple retention rates (e.g., 66%, 49%, and 33%) in Table 4. The reported metrics include end-to-end inference time, prefill runtime, GPU memory consumption (in MB), retained token count, and speedup ratios relative to the baseline.

| Method | Token (%) | Total Time (Min:Sec) | Prefill Time (Min:Sec) | TFLOPs | Cuda Memory (MB) | Speedup (Total) | (Prefill) |
|---|---|---|---|---|---|---|---|
| Qwen2.5-VL-7B | 100% | 67:03 | 66:04 | 930.65 | 34791 | 1.00x | 1.00x |
| DART | 66% | 52:17 | 51:51 | 715.67 | 29134 | 1.28x | 1.27x |
| ToME | 66% | 48:38 | 47:44 | 715.63 | 29134 | 1.38x | 1.38x |
| **TokenSculpt** | 66% | 49:31 | 48:59 | 715.63 | 29134 | 1.35x | 1.35x |
| DART | 49% | 43:54 | 43:24 | 608.09 | 26302 | 1.53x | 1.52x |
| ToME | 49% | 41:27 | 40:44 | 608.12 | 26303 | 1.62x | 1.62x |
| **TokenSculpt** | 49% | 42:21 | 41:28 | 608.12 | 26303 | 1.58x | 1.59x |
| DART | 33% | 37:00 | 36:30 | 506.93 | 24936 | 1.81x | 1.81x |
| ToME | 33% | 35:11 | 34:39 | 506.96 | 24274 | 1.90x | 1.90x |
| **TokenSculpt** | 33% | 35:43 | 35:18 | 506.93 | 24274 | 1.88x | 1.87x |

Table 4: Efficiency analysis of **TokenSculpt** on Qwen2.5-VL 7B with 66%, 49%, 33% of tokens retained.

## F  INFLUENCE FROM SAMPLING STRATEGIES.

We provide a comprehensive comparison of different token sampling strategies under varying retention rates (e.g., 66%, 49%, and 33%) in Table 5.

| Retention | Method | Charades-STA | | | |
|---|---|---|---|---|---|
| | | R@0.3 | R@0.5 | R@0.7 | mIoU |
| **100%** | baseline | 66.59 | 46.26 | 22.47 | 43.70 |
| **66%** | uniform | 64.65 | 44.73 | 20.83 | 42.31 |
| | random | 64.54 | 44.75 | 21.37 | 42.34 |
| | pivot | 61.24 | 41.24 | 20.49 | 40.54 |
| **49%** | uniform | 63.68 | 43.55 | 20.27 | 41.63 |
| | random | 62.55 | 43.03 | 20.13 | 41.24 |
| | pivot | 55.86 | 37.55 | 17.96 | 37.24 |
| **33%** | uniform | 59.65 | 39.95 | 19.36 | 39.52 |
| | random | 58.60 | 38.60 | 17.66 | 38.45 |
| | pivot | 50.32 | 33.11 | 16.07 | 33.66 |

Table 5: Sample strategy analysis of **TokenSculpt** on Qwen2.5-VL 3B with 66%, 49%, 33% of tokens retained.

## G  EVIDENCE FOR DIVERSE SPATIOTEMPORAL STRUCTURE PRESERVATION.

**Min–Max Anchors Preserve Boundaries.**  We provide the formal definition of the min–max duplication strategy in Section 3.2. Through anchor-based token selection, our method ensures that tokens with high similarity remain confined within a compact spatial cube, effectively preserving their inherent spatio-temporal locality.

**Limitations of Traditional Token Pruning Methods.**  As discussed in Section 3.3, conventional token pruning approaches exhibit two major drawbacks. Attention-based pruning suffers from bottom bias: the attention distribution tends to over-focus on semantic centers while neglecting boundary tokens, leading to the loss of structural edges during pruning. Similarity-based clustering methods may merge semantically similar yet spatially disjoint regions, resulting in the so-called

huge-cluster phenomenon that disrupts the original spatiotemporal topology. Motivated by these observations, our proposed **TokenSculpt** explicitly retains extremal boundary tokens, avoiding both bottom-bias and huge-cluster issues, thereby maintaining the global structure after pruning.

**Task-Level Evidence: Improved Performance on Grounding.** Grounding tasks are highly sensitive to structural completeness. Under a 33% retention ratio, **TokenSculpt** achieves a +3.6% mIoU improvement over ToMe on Charades-STA and ActivityNet-Grounding, demonstrating superior preservation of structural information.

**Visualization: Clear Preservation of Structural Boundaries.** As illustrated in Figure 2, **TokenSculpt** preserves the panda's white head boundary. The visual results further verify that the proposed Min–Max anchor strategy effectively maintains true spatial topology.

