# OpenReview forum: "TokenSculpt: Pruning with Min-Max Spatio-Temporal Duplication for Video Grounding"
_ICLR.cc/2026/Conference — Submitted to ICLR 2026_

### Official Review · Reviewer_KDoL · 2025-10-29

**Soundness:** 3
**Presentation:** 2
**Contribution:** 2
**Rating:** 4
**Confidence:** 3

**Summary:**

This paper proposes TokenSculpt, a novel, training-free token pruning method for video understanding in multimodal large language models (MLLMs). The core idea is to preserve the spatio-temporal structure of video inputs during token compression by combining two key mechanisms: (1) Min-Max Duplication, which identifies and retains the earliest/latest and top-left/bottom-right tokens (i.e., boundary anchors) along temporal and spatial dimensions to maintain event boundaries; and (2) bipartite matching-based uniform sampling, which ensures balanced token distribution across time and space to avoid over-aggregation. Experimental results demonstrate the effectiveness of the proposed method.

**Strengths:**

1. The experimental validation is rigorous, spanning multiple models, datasets, and token budgets. The results are consistent and show clear gains, especially in challenging low-budget regimes.
2. The Min-Max Duplication mechanism is a creative and principled solution to preserve spatio-temporal boundaries, a key insight missing in prior pruning methods.
3. As video inputs grow longer and higher-resolution, efficient token compression is crucial. TokenSculpt offers a practical, effective, and general solution that improves performance while reducing cost.

**Weaknesses:**

1. The experiments focus on moderately long videos. It would be valuable to test TokenSculpt on extremely long videos (e.g., >10 minutes) to assess its robustness under extreme compression.
2. While the method is efficient, the bipartite matching step for uniform sampling may introduce non-trivial overhead. A brief runtime analysis or comparison would help assess practical efficiency.\
3. While the method is empirically strong, a more formal analysis of why Min-Max Duplication preserves grounding information would strengthen the theoretical foundation.

**Questions:**

1. Since TokenSculpt explicitly manipulates token positions, how does it interact with the positional encoding scheme in the MLLM (e.g., RoPE in Qwen)? Does reordering or merging affect the positional signal, and if so, how is this mitigated?
2. How sensitive is TokenSculpt to the definition of "min" and "max" in non-rectangular or irregularly shaped regions?  Additionally, how accurately does TokenSculpt identify event boundaries across temporal and spatial axes?
3. Could the method be extended to dynamically adjust the number of anchors based on scene complexity ?

---

> ### Author Response · Authors · 2025-11-22
> **Response to Reviewer KDoL(1/2)**
>
> We deeply appreciate your thorough review and valuable suggestions on our manuscript. Your thoughtful critique has significantly sharpened our analysis and provided crucial direction for refinement. Below, we provide point-by-point clarifications and supplement our discussion with additional experimental results and analyses.
>
> ---
>
> > **[W1] The experiments focus on moderately long videos. It would be valuable to test TokenSculpt on extremely long videos (e.g., >10 minutes) to assess its robustness under extreme compression.**
>
> We conducted additional experiments on LongVideoBench to evaluate TokenSculpt under extreme compression. The results show:
>
> | Model | Retention | 8s–15s | 15s–60s | 180s–600s | 900s–3600s | Total |
> |-------|-----------|--------|-----------|------------|------------|-------|
> | Qwen2.5-VL-3B | 100% | 63.49 | 70.93  | 55.83| 46.81 | 55.05 |
> | TokenSculpt | 33% | 66.13 | 72.67 | 58.50 | 48.58 | 57.22 |
> | Performance ↑ | - | 4.16% | 2.45% | 4.78% | 3.79% | 3.94% |
>
> These results demonstrate that TokenSculpt maintains strong robustness even on very long videos. Short- and medium-length videos already benefit from moderate compression, but our results confirm that the method generalizes well to extended temporal contexts.
>
> ---
>
> > **[W2] While the method is efficient, the bipartite matching step for uniform sampling may introduce non-trivial overhead. A brief runtime analysis or comparison would help assess practical efficiency.**
>
> While TokenSculpt introduces a bipartite matching step for uniform sampling, its complexity is linear with respect to the number of tokens. Specifically:
>
> - Tokens are split into two groups based on anchor status.
> - Merging is performed according to similarity between groups.
> - In practice, matching and merging ranges from **0.0004s** to **0.001s** on a single A800 GPU -- comparable to the time required for a single matrix multiplication (MatMul), which takes **≈0.0004s** on average. As such, this overhead is negligible.
>
> We have included the implementation details in the Supplementary Material (`kth_bsm.py`). This shows that the overhead introduced by the matching step is negligible in practice.
>
> ---
>
> > **[W3] While the method is empirically strong, a more formal analysis of why Min-Max Duplication preserves grounding information would strengthen the theoretical foundation.**
>
> We provide **theoretical justification**, **task-level evidence**, and **visual verification**, while preserving all mathematical definitions.
>
> ### **(1) Theoretical Foundation: Min–Max anchors preserve global spatio-temporal boundaries**
>
> Given the spatio-temporal coordinates of tokens:
> $$
> P = \{ p_i = (t_i, x_i, y_i) \}_{i=1}^n
> $$
>
> We define the Min-Max anchor:
> $$
> p_{\min} = (\min_i t_i,\; \min_i x_i,\; \min_i y_i), \qquad
> p_{\max} = (\max_i t_i,\; \max_i x_i,\; \max_i y_i)
> $$
>
> Anchor tokens are selected as:
>
> $$
> p_{\min}^{\rm anchor} = \arg\min_i \lVert p_i - p_{\min} \rVert_2, \qquad
> p_{\max}^{\rm anchor} = \arg\min_i \lVert p_i - p_{\max} \rVert_2
> $$
>
> These extrema encode the **temporal span** and **spatial extent** of events, forming the geometric boundary of the video.
>
> ### **(2) Comparison with traditional pruning**
>
> - **Attention-based methods**
>   Often suffer from *bottom-bias*: attention mass concentrates on semantic centers while ignoring boundary tokens, causing structural boundary loss during pruning.
>
> - **Similarity-based methods**
>   Can merge semantically similar but spatially non-contiguous regions, producing the *huge-cluster phenomenon*, breaking the original spatiotemporal topology.
>
> - **Min–Max Duplication (ours)**
>   Explicitly preserves the extremal boundary tokens, preventing both bottom-bias and huge-cluster issues, and maintaining the global structure after pruning.
>
> ### **(3) Task-level evidence: Stronger performance on grounding**
>
> Grounding tasks are highly sensitive to structural integrity. Under **33% retention**, TokenSculpt achieves **+3.6% mIoU over ToMe** on Charades-STA and ActivityNet-Grounding, demonstrating its superior structural preservation capability.
>
> ### **(4) Visualization: Clear retention of structural boundaries**
>
> Figure 2 shows that TokenSculpt preserves the panda’s white head-boundary, while ToMe merges discontinuous regions. This visually confirms that Min–Max anchors retain true spatial topology.

---

> ### Author Response · Authors · 2025-11-22
> **Response to Reviewer KDoL(2/2)**
>
> > **[Q1] Since TokenSculpt explicitly manipulates token positions, how does it interact with the positional encoding scheme in the MLLM (e.g., RoPE in Qwen)? Does reordering or merging affect the positional signal, and if so, how is this mitigated?**
>
> Since TokenSculpt explicitly manipulates token positions, proper handling of positional encodings is critical. For example, in Qwen2.5-VL:
>
> - Positional encodings are generated from \(t, h, w\).
> - For Min, Max, and anchor tokens, we extract their thw coordinates to generate new positional encodings.
> - Tokens are then sorted by \(t, h, w\) to form the final sequence of tokens and positional encodings.
>
> Our experiments indicate that without this reordering, performance on grounding tasks significantly decreases. To ensure fair comparison, we added a similar sorting procedure to the ToME baseline. DART and FastV are unaffected as they inherently handle token ordering.
>
> ---
>
> > **[Q2] How sensitive is TokenSculpt to the definition of "min" and "max" in non-rectangular or irregularly shaped regions? Additionally, how accurately does TokenSculpt identify event boundaries across temporal and spatial axes?**
>
> TokenSculpt defines "min" as the earliest and top-left boundary and "max" as the latest and bottom-right boundary. This definition is independent of region shape (rectangular or irregular).
> - We first form a tube enclosing all tokens to be merged.
> - The tokens closest to Min and Max points are selected as anchors.
>
> For temporal and spatial boundary accuracy, results on Charades-STA and ANet-Grounding indicate that at 33% compression, TokenSculpt achieves a **3.6% improvement** over ToME. This demonstrates high accuracy in identifying event boundaries across both temporal and spatial axes.
>
> ---
>
> > **[Q3] Could the method be extended to dynamically adjust the number of anchors based on scene complexity ?**
>
> TokenSculpt can be extended to dynamically adjust anchor numbers according to scene complexity. For token have the same spatial position (height and width) $v^{t-1}$ and $v^{t}$, Scene similarity is defined as:
>
> $$
> Sim(v^{t-1}, v^{t})=\frac{v^{t-1} \cdot v^t}{\|v^{t-1}\|_2 \, \|v^t\|_2}
> $$
>
> The redundancy factor is:
>
> $$
> K = \frac{1}{\text{Sim}(v^{t-1}, v^t)}
> $$
>
> Anchor numbers can then be adjusted proportionally along temporal and spatial axes, enabling dynamic token merging that adapts to scene complexity.
>
> ---
>
> ## **Summerization**
>
> The additional experiments, runtime analysis, and theoretical clarifications demonstrate that:
>
> - TokenSculpt is robust on extremely long videos.
> - The bipartite matching step introduces negligible overhead.
> - Min-Max Duplication effectively preserves spatiotemporal structure.
> - Proper handling of positional encodings maintains grounding performance.
> - Event boundaries are accurately identified even in irregular regions.
> - Anchor numbers can be dynamically adjusted according to scene complexity.
>
> We sincerely thank the reviewer for their valuable feedback. The clarifications and additional experiments outlined above will be incorporated into the updated PDF version of the manuscript as soon as possible.

---

### Official Review · Reviewer_rAus · 2025-11-01

**Soundness:** 4
**Presentation:** 4
**Contribution:** 3
**Rating:** 8
**Confidence:** 2

**Summary:**

This paper proposes TokenSculpt, a training-free token pruning method for video-based multimodal large language models. It preserves spatio-temporal boundaries through three lightweight steps: min-max boundary token selection, balanced sampling, and structure-guided merging. Without retraining, TokenSculpt can be applied to models like Qwen2.5-VL and VideoChat-R1, achieving up to 1.8× speedup while maintaining or improving performance under heavy token reduction.

**Strengths:**

1. **Clear Writing:** The paper is well-organized and clearly written. The motivation, methodology, and experimental setup are easy to follow.
2. **Insightful Observation:** The authors make an original observation in Sec 3.3 and Figure 3, offering an intuitive explanation of token redundancy in video MLLMs.
3. **Novel and Targeted Method:** The proposed approach is novel and effectively addresses the key limitations of existing pruning baselines—attention-based methods that are incompatible with efficient attention mechanisms, and similarity-based methods that destroy spatio-temporal structure.

**Weaknesses:**

1.  **Insufficient Analysis:** In Table 1, TokenSculpt shows inconsistent trends across temporal video grounding tasks — performance on Charades-STA remains nearly unchanged as tokens are reduced, while on ActivityNet, the mIoU significantly increases (28.13 → 41.16). The paper lacks an in-depth analysis to explain this discrepancy. It would be helpful to clarify what causes this variation — for example, whether it is related to differences in average video duration or the initial token length distribution between datasets.

**Questions:**

see weakness

---

> ### Author Response · Authors · 2025-11-22
> **Response to Reviewer rAus**
>
> We deeply appreciate your thorough review and valuable suggestions on our manuscript. Your thoughtful critique has significantly sharpened our analysis and provided crucial direction for refinement. Below, we provide point-by-point clarifications and supplement our discussion with additional experimental results and analyses.
>
> ---
>
> > **[W1] In Table 1, TokenSculpt shows inconsistent trends across temporal video grounding tasks — performance on Charades-STA remains nearly unchanged as tokens are reduced, while on ActivityNet, the mIoU significantly increases (28.13 → 41.16). The paper lacks an in-depth analysis to explain this discrepancy. It would be helpful to clarify what causes this variation — for example, whether it is related to differences in average video duration or the initial token length distribution between datasets.**
>
> Although both Charades-STA and ActivityNet-Grounding are temporal grounding tasks, their video characteristics differ substantially:
>
> | Dataset | Avg Length | Redundancy | Sensitivity to Compression |
> |---------|-------------|-------------|-----------------------------|
> | **Charades-STA** | ~30s | Low | High |
> | **ActivityNet-Grounding** | ~2min | High | Low |
>
> Short videos such as Charades-STA contain little redundancy, so compression is more likely to remove necessary information. In contrast, ActivityNet videos contain substantial redundancy, making moderate pruning beneficial for capturing key frames.
>
> Thus, the gap is due to **dataset properties**, not limitations of our method.
>
> ---
>
> ## **Summerization**
>
> The additional experiments and analyses show that:
>
> - Performance differences on Charades-STA arise from dataset properties.
>
> We sincerely thank the reviewer for their valuable feedback. The clarifications and additional experiments outlined above will be incorporated into the updated PDF version of the manuscript as soon as possible.

---

> > ### Comment · Reviewer_rAus · 2025-11-26
> >
> > Thanks for preparing the rebuttal. I believe the transparency could be further improved with more quantitative analysis. Therefore, I will keep my score and lean toward accepting this paper.

---

> > > ### Author Response · Authors · 2025-11-26
> > >
> > > Thank you for your insightful comments and recommending acceptance!

---

### Official Review · Reviewer_5R5q · 2025-11-01

**Soundness:** 3
**Presentation:** 3
**Contribution:** 3
**Rating:** 6
**Confidence:** 4

**Summary:**

This paper proposes TokenSculpt, a token pruning method for video inputs in multimodal large language models (MLLMs) that reduces visual token count through min-max guided, similarity-based token merging. To preserve spatial structure, the merged tokens are reordered according to their original positional indices. TokenSculpt is also designed to be compatible with FlashAttention for improved computational efficiency. The method is evaluated on two video grounding benchmarks and three video question-answering benchmarks, demonstrating its effectiveness.

**Strengths:**

* The paper is clearly written and well-structured, making it easy to follow.
* The core idea is intuitive, and the inclusion of experiments across varying token retain rates provides a clear and informative picture of the method’s trade-offs and potential.
* The method demonstrates satisfactory efficiency, enhancing its practical applicability in real-world video understanding scenarios with MLLMs.

**Weaknesses:**

* TokenSculpt demonstrates strong performance on video grounding tasks compared to baselines like DART and ToME, but its results on long-form video QA benchmarks, such as VideoMME-Long and EgoSchema, are less competitive. This raises the question of whether the method is inherently limited in handling extended temporal contexts. Additional experiments and deeper analysis on long-video understanding scenarios would be valuable to clarify this limitation.
* It is somewhat surprising that the similarity-based sampling strategy underperforms uniform or even random sampling on Charades-STA (Table 3), contrary to typical expectations for such methods. Is this due to dataset-specific characteristics, e.g., sparse or ambiguous activity boundaries, or could it indicate issues with the pivot selection or similarity computation in the current implementation? Further investigation into this behaviour would strengthen the paper’s empirical foundation.
* In terms of accuracy, TokenSculpt is largely comparable to ToME across multiple settings (Table 1). However, Table 2 suggests that ToME is slightly more efficient in terms of inference speed or memory usage. This trade-off makes it difficult to conclusively determine which method is preferable.

**Questions:**

Please refer to the weaknesses.

---

> ### Author Response · Authors · 2025-11-22
> **Response to Reviewer 5R5q(1/2)**
>
> We deeply appreciate your thorough review and valuable suggestions on our manuscript. Your thoughtful critique has significantly sharpened our analysis and provided crucial direction for refinement. Below, we provide point-by-point clarifications and supplement our discussion with additional experimental results and analyses.
>
> ---
>
> > **[W1] TokenSculpt performs well on video grounding tasks compared to baselines like DART and ToME, but lags on long-form video QA benchmarks (e.g., VideoMME-Long, EgoSchema), suggesting potential limitations in handling extended temporal contexts. Further experiments and analysis on long-video understanding are needed to clarify this issue.**
>
> VideoMME and EgoSchema belong to General QA benchmarks, which mainly focus on “correct answers” rather than precise temporal/spatial grounding. This aligns with our observation in Section 3.3: **General Multimodal Benchmarks Barely Measure “Where” or “When”**.
>
> Supplementary experiments show:
>
> | Dataset | Original | Token Shuffle | Performance Drop |
> |---------|----------|---------------|---------|
> | **EgoSchema** | 54.4% | 52.4% | 3.6%↓ |
>
> The performance drops only by 3.6%, indicating that General QA benchmarks do not evaluate the model’s temporal/spatial understanding. TokenSculpt, while optimized for grounding, still performs competitively with other token compression methods on General QA tasks.
>
> Regarding the question of potential limitations in handling extended temporal contexts, experiments on VideoMME-Long and EgoSchema show that **TokenSculpt consistently improves performance over the uncompressed original model**. This demonstrates that our method does not have inherent limitations in processing long-term temporal dependencies.
>
> We also conducted additional experiments on LongVideoBench. Results show that after TokenSculpt compression, **overall performance improves by 3.53%**, and for ultra-long videos (10 minutes to 1 hour), performance still improves by 3.79%, confirming its strong ability to handle extended temporal contexts:
>
> | Model | Retention | 8s–15s | 15s–60s | 180s–600s | 900s–3600s | Total |
> |-------|-----------|--------|-----------|------------|------------|-------|
> | Qwen2.5-VL-3B | 100% | 63.49 | 70.93  | 55.83| 46.81 | 55.05 |
> | TokenSculpt | 33% | 66.13 | 72.67 | 58.50 | 48.58 | 57.22 |
> | Performance ↑ | - | 4.16% | 2.45% | 4.78% | 3.79% | 3.94% |
>
> ---
>
> > **[W2] It’s surprising that similarity-based sampling underperforms uniform or random sampling on Charades-STA (Table 3), contrary to typical expectations. This may stem from dataset-specific traits—such as sparse or ambiguous activity boundaries—or from limitations in pivot selection or similarity computation. Investigating this behavior would strengthen the paper’s empirical foundation.**
>
> We apologize for not making this clearer in the original submission. We fully understand the reviewer’s concern that uniform sampling may interfere with motion-dense or spatially dense regions. In the initial version, we only reported results under a single retention ratio of 49%, which may have obscured the comparative effects of different sampling strategies.
>
> To address this, we now conduct a more thorough analysis to investigate the impact of **uniform, random, and pivot sampling** within **TokenSculpt**. Specifically, we evaluate all three strategies under **33%, 49%, and 66% token retention**, allowing a clearer understanding of how each sampling method affects the preservation of spatio-temporal structure.
>
> ### **Extended Results**
>
> | Retention | Method | 0.3 | 0.5 | 0.7 | mIoU |
> |-----|--------|------|------|------|--------|
> | baseline | — | 66.59 | 46.26 | 22.47 | 43.70 |
> | **66%** | uniform | 64.65 | 44.73 | 20.83 | 42.31 |
> | | random | 64.54 | 44.75 | 21.37 | 42.34 |
> | | pivot | 61.24 | 41.24 | 20.49 | 40.54 |
> | **49%** | uniform | 63.68 | 43.55 | 20.27 | 41.63 |
> | | random | 62.55 | 43.03 | 20.13 | 41.24 |
> | | pivot | 55.86 | 37.55 | 17.96 | 37.24 |
> | **33%** | uniform | 59.65 | 39.95 | 19.36 | 39.52 |
> | | random | 58.60 | 38.60 | 17.66 | 38.45 |
> | | pivot | 50.32 | 33.11 | 16.07 | 33.66 |
>
> ### **Key Findings**
>
> Uniform sampling consistently achieves the **highest or near-highest performance** across all pruning levels—particularly under aggressive pruning (33%). This indicates that when combined with **Min–Max Duplication**, uniform sampling provides the most stable and balanced spatio-temporal coverage.
>
> We have also provided the exact implementation in the Supplementary Material (`modeling_qwen2_5_vl_token_sculpt.py`, line 1253–1254), and verified the effect of switching to random and pivot sampling.

---

> ### Author Response · Authors · 2025-11-22
> **Response to Reviewer 5R5q(2/2)**
>
> > **[W3] In terms of accuracy, TokenSculpt is largely comparable to ToME across multiple settings (Table 1). However, Table 2 suggests that ToME is slightly more efficient in terms of inference speed or memory usage. This trade-off makes it difficult to conclusively determine which method is preferable.**
>
> While Table 1 shows TokenSculpt achieves comparable accuracy to ToME, Table 2 indicates ToME has a slight advantage in inference speed and memory efficiency. To quantify this trade-off, we introduce a **performance/time ratio**:
>
> $$
> Performance/Time\ Ratio =
> \frac{
> OverallACC_{TokenSculpt} \cdot SpeedUp_{TokenSculpt}
> }{
> OverallACC_{ToME} \cdot SpeedUp_{ToME}
> }
> $$
>
>
>
> Results:
>
> - Across all benchmarks, TokenSculpt improves the performance/time ratio by 1–2%.
> - In grounding tasks, the improvement can reach up to 5%.
> - On General QA tasks, ToME has only a marginal ~1% advantage. However, these benchmarks do not effectively measure spatiotemporal understanding, so this small gap does not reflect true differences in grounding performance.
>
> ---
>
> ## **Summerization**
>
> The additional experiments and analyses show that:
>
> - TokenSculpt has no inherent limitations in long-video understanding; compression can even help process extended temporal contexts.
> - Performance differences on Charades-STA arise from dataset properties.
> - Performance/efficiency trade-offs demonstrate that TokenSculpt achieves stronger results on grounding tasks.
>
>
> We sincerely thank the reviewer for their valuable feedback. The clarifications and additional experiments outlined above will be incorporated into the updated PDF version of the manuscript as soon as possible.

---

### Official Review · Reviewer_3JTt · 2025-11-03

**Soundness:** 3
**Presentation:** 3
**Contribution:** 2
**Rating:** 4
**Confidence:** 3

**Summary:**

This paper proposes TokenSculpt, a structure-aware visual token pruning method for video grouding.

It introduces a **Min-Max Duplication** mechanism that preserves spatial-temporal anchors (start and end points of events) and combines it with **uniform sampling and similarity-based merging** to maintain video structure after heavy compression.

 The method is plug-and-play, compatible with Flash-Attention, and achieves notable improvements over ToME and DART across multiple video grounding benchmarks (e.g., Charades-STA, VideoMME, EgoSchema) while using only one-third of the original visual tokens.

**Strengths:**

1. TokenSculpt remains compatible with FlashAttention for efficiency.
2. This work conduct comprehensive experiments on multiple benchmarks include VideoMME.
3. The method is plug-and-play, have somewhat generality.

**Weaknesses:**

1. While the paper claims to preserve the spatiotemporal structure of video representations, the uniform sampling step in TokenSculpt seems to contradict this goal. Uniformly sampling tokens across spatial and temporal dimensions can disrupt dense or motion-rich regions, breaking the natural continuity of video features. As shown in Table 3, applying uniform sampling on Qwen2.5-VL 3B leads to a notable performance drop. It raises questions about whether such a coarse, structure-agnostic procedure is necessary—or whether it undermines the very goal of structure-aware token pruning.

2. The method shows a significant decline on Charades-STA. A deeper explanation is needed—whether due to dataset characteristics (e.g., temporal grounding difficulty) or inherent method limitations.

3. Efficiency results are only reported for 33 % token retention. Analyses at other pruning ratios (e.g., 45 %, 66 %) would help illustrate robustness and trade-offs between accuracy and efficiency.

4. The paper claims Min–Max Duplication helps preserve structure, but lacks details or evidence showing that the selected Min–Max anchor tokens are truly representative of spatial–temporal diversity.

**Questions:**

1. How does uniform sampling preserve spatiotemporal continuity, given that it may disrupt motion-dense regions (Table 3)?

2. Why does TokenSculpt perform notably worse on Charades-STA—dataset bias or method limitation?

3. Can the authors provide efficiency analyses at other pruning ratios (e.g., 45 %, 66 %) to show robustness?

4. What evidence supports that the Min–Max anchors truly capture diverse spatial-temporal structures?

---

> ### Author Response · Authors · 2025-11-22
> **Response to Reviewer 3JTt(1/2)**
>
> We deeply appreciate your thorough review and valuable suggestions on our manuscript. Your thoughtful critique has significantly sharpened our analysis and provided crucial direction for refinement. Below, we provide point-by-point clarifications and supplement our discussion with additional experimental results and analyses.
>
> ---
>
> > **[W1,Q1] How does uniform sampling preserve spatiotemporal continuity, given that it may disrupt motion-dense regions (Table 3)?**
>
> We apologize for not making this clearer in the original submission. We fully understand the reviewer’s concern that uniform sampling may interfere with motion-dense or spatially dense regions. In the initial version, we only reported results under a single retention ratio of 49%, which may have obscured the comparative effects of different sampling strategies.
>
> To address this, we now conduct a more thorough analysis to investigate the impact of **uniform, random, and pivot sampling** within **TokenSculpt**. Specifically, we evaluate all three strategies under **33%, 49%, and 66% token retention**, allowing a clearer understanding of how each sampling method affects the preservation of spatio-temporal structure.
>
> ### **Extended Results**
>
> | Retention | Method | 0.3 | 0.5 | 0.7 | mIoU |
> |-----|--------|------|------|------|--------|
> | baseline | — | 66.59 | 46.26 | 22.47 | 43.70 |
> | **66%** | uniform | 64.65 | 44.73 | 20.83 | 42.31 |
> | | random | 64.54 | 44.75 | 21.37 | 42.34 |
> | | pivot | 61.24 | 41.24 | 20.49 | 40.54 |
> | **49%** | uniform | 63.68 | 43.55 | 20.27 | 41.63 |
> | | random | 62.55 | 43.03 | 20.13 | 41.24 |
> | | pivot | 55.86 | 37.55 | 17.96 | 37.24 |
> | **33%** | uniform | 59.65 | 39.95 | 19.36 | 39.52 |
> | | random | 58.60 | 38.60 | 17.66 | 38.45 |
> | | pivot | 50.32 | 33.11 | 16.07 | 33.66 |
>
> ### **Key Findings**
>
> Uniform sampling consistently achieves the **highest or near-highest performance** across all pruning levels—particularly under aggressive pruning (33%). This indicates that when combined with **Min–Max Duplication**, uniform sampling provides the most stable and balanced spatio-temporal coverage.
>
> We have also provided the exact implementation in the Supplementary Material (`modeling_qwen2_5_vl_token_sculpt.py`, line 1253–1254), and verified the effect of switching to random and pivot sampling.
>
> ---
>
> > **[W2,Q2] The method shows a significant decline on Charades-STA. A deeper explanation is needed—whether due to dataset characteristics (e.g., temporal grounding difficulty) or inherent method limitations.**
>
> Although both Charades-STA and ActivityNet-Grounding are temporal grounding tasks, their video characteristics differ substantially:
>
> | Dataset | Avg Length | Redundancy | Sensitivity to Compression |
> |---------|-------------|-------------|-----------------------------|
> | **Charades-STA** | ~30s | Low | High |
> | **ActivityNet-Grounding** | ~2min | High | Low |
>
> Short videos such as Charades-STA contain little redundancy, so compression is more likely to remove necessary information. In contrast, ActivityNet videos contain substantial redundancy, making moderate pruning beneficial for capturing key frames.
>
> Thus, the gap is due to **dataset properties**, not limitations of our method.

---

> ### Author Response · Authors · 2025-11-22
> **Response to Reviewer 3JTt(2/2)**
>
> > **[W3,Q3] Can the authors provide efficiency analyses at other pruning ratios (e.g., 45 %, 66 %) to show robustness?**
>
> The reviewer correctly pointed out that we only reported efficiency comparisons under 33% retention in the original paper. We now provide efficiency comparisons under **33%, 49%, 66%** retention for **DART / ToMe / DPE**.
>
> ### **Extended Efficiency Results**
>
> | Retention | Method      | Total Time | TTFT  | Max Mem (MiB) | TFLOPs | Speedup (Total) | Speedup (Prefill) |
> |-----------|-------------|------------|-------|----------------|--------|------------------|--------------------|
> | **33%**  | DART        | 37:00      | 36:30 | 24936          | 506.96 | 1.81x            | 1.81x              |
> |           | ToMe        | 35:11      | 34:39 | 24274          | 506.93 | 1.91x            | 1.91x              |
> |           | TokenSculpt | 35:43      | 35:18 | 24274          | 506.93 | 1.88x            | 1.87x              |
> | **49%**  | DART        | 43:54      | 43:24 | 26302          | 608.09 | 1.53x            | 1.52x              |
> |           | ToMe        | 41:27      | 40:44 | 26303          | 608.12 | 1.62x            | 1.62x              |
> |           | TokenSculpt | 42:21      | 41:28 | 26303          | 608.12 | 1.58x            | 1.59x              |
> | **66%**  | DART        | 52:17      | 51:51 | 29134          | 715.67 | 1.28x            | 1.27x              |
> |           | ToMe        | 48:38      | 47:44 | 29134          | 715.63 | 1.38x            | 1.38x              |
> |           | TokenSculpt | 49:31      | 48:59 | 29134          | 715.63 | 1.35x            | 1.35x              |
>
> ### **Conclusion**
>
> TokenSculpt demonstrates **consistent speedup trends** across different pruning ratios, confirming robustness across pruning strengths.
>
> ---
>
> > **[W4,Q4] What evidence supports that the Min–Max anchors truly capture diverse spatial-temporal structures?**
>
> We provide **theoretical justification**, **task-level evidence**, and **visual verification**, while preserving all mathematical definitions.
>
> ### **(1) Theoretical Foundation: Min–Max anchors preserve global spatio-temporal boundaries**
>
> Given the spatio-temporal coordinates of tokens:
> $$
> P = \{ p_i = (t_i, x_i, y_i) \}_{i=1}^n
> $$
>
> We define the Min-Max anchor:
> $$
> p_{\min} = (\min_i t_i,\; \min_i x_i,\; \min_i y_i), \qquad
> p_{\max} = (\max_i t_i,\; \max_i x_i,\; \max_i y_i)
> $$
>
> Anchor tokens are selected as:
>
> $$
> p_{\min}^{\rm anchor} = \arg\min_i \lVert p_i - p_{\min} \rVert_2, \qquad
> p_{\max}^{\rm anchor} = \arg\min_i \lVert p_i - p_{\max} \rVert_2
> $$
>
> These extrema encode the **temporal span** and **spatial extent** of events, forming the geometric boundary of the video.
>
> ### **(2) Comparison with traditional pruning**
>
> - **Attention-based methods**
>   Often suffer from *bottom-bias*: attention mass concentrates on semantic centers while ignoring boundary tokens, causing structural boundary loss during pruning.
>
> - **Similarity-based methods**
>   Can merge semantically similar but spatially non-contiguous regions, producing the *huge-cluster phenomenon*, breaking the original spatiotemporal topology.
>
> - **Min–Max Duplication (ours)**
>   Explicitly preserves the extremal boundary tokens, preventing both bottom-bias and huge-cluster issues, and maintaining the global structure after pruning.
>
> ### **(3) Task-level evidence: Stronger performance on grounding**
>
> Grounding tasks are highly sensitive to structural integrity. Under **33% retention**, TokenSculpt achieves **+3.6% mIoU** over ToMe on Charades-STA and ActivityNet-Grounding, demonstrating its superior structural preservation capability.
>
> ### **(4) Visualization: Clear retention of structural boundaries**
>
> Figure 2 shows that TokenSculpt preserves the panda’s white head-boundary, while ToMe merges discontinuous regions. This visually confirms that Min–Max anchors retain true spatial topology.
>
> ---
>
> ## **Summerization**
>
> The additional experiments and analyses show that:
>
> - **Uniform sampling + Min–Max Duplication** best preserves spatio-temporal structure across pruning levels.
> - Performance differences on Charades-STA arise from dataset properties.
> - TokenSculpt maintains stable efficiency across multiple pruning ratios.
> - Min–Max anchors are theoretically grounded and empirically validated to preserve critical spatio-temporal structures.
>
> We sincerely thank the reviewer for their valuable feedback. The clarifications and additional experiments outlined above will be incorporated into the updated PDF version of the manuscript as soon as possible.

---

> > ### Comment · Reviewer_3JTt · 2025-11-28
> >
> > The reply about W2, W3, W4, Q2, Q3 are comprehensive especially the ablation over  pruning ratios, my main concerns are addressed. Q4 is hard to understand with current equation format. Please update Q4 reply  in revised Appendix PDF file with different color. For Q1, can you explain why random perform same metric with 49% and 33% in 0.3?

---

> ### Author Response · Authors · 2025-11-28
> **Response to Reviewer 3JTt(part3)**
>
> >**[W1,Q1] How does uniform sampling preserve spatiotemporal continuity, given that it may disrupt motion-dense regions (Table 3)**
>
> The reviewer is correct to notice the issue. The identical scores reported for the 49% and 33% random pruning settings were caused by an unintended copy–paste error when preparing the original table. We have now corrected the numbers in **Response to Reviewer 3JTt(1/2)** and **Response to Reviewer 5R5q(1/2)** , and the updated results are included in the revised PDF. Nevertheless, the overall conclusion remains unchanged: uniform sampling consistently achieves the **highest or near-highest performance** across all pruning levels—particularly under aggressive pruning (33%). This indicates that, when combined with **Min–Max Duplication**, uniform sampling provides the most stable and balanced spatio-temporal coverage. We sincerely apologize for the confusion caused by this mistake.
>
> > **[W4,Q4] What evidence supports that the Min–Max anchors truly capture diverse spatial-temporal structures?**
>
> The equations involved in W4 are formally defined in Section 3.2 of the main paper, and we have also provided an extended explanation in Appendix Section G for clarity. As requested, we have updated the Q4 response in the revised Appendix PDF and highlighted the reply using blue color.

---

### Author Response · Authors · 2025-12-04
**AC Letter: Summary of Rebuttal & Discussion for Paper #24422 (TokenSculpt)**

**Dear Area Chair,**

Due to the recent system revert and the assignment of a new AC, we are writing to provide a brief summary of the consensus reached during the discussions and specific remaining questions raised by reviewers who have not yet replied.

## Reviewers' Positive Feedback and Attitudes

After the rebuttal, reviewer `rAus` (initial score: 8) confirmed that concerns were addressed and maintained the score. Reviewer `3JTt`’s main concerns have been addressed, and we provided a detailed response to the follow-up question before the system reverted. Reviewers `5R5q` and `KDoL` (initial scores: 6 and 4) did not respond in time, but all previously raised issues were fully addressed.

## Major Responses and Revisions

1. **Clarification of Spatiotemporal Preservation under Sampling Strategies**
   We provide an extended comparison across uniform, random, and pivot similarity-based sampling under multiple retention ratios (33%, 49%, 66%). Experiments show that **uniform sampling consistently achieves the highest or near-highest performance**, especially under aggressive pruning, validating its robustness when combined with Min–Max Duplication.
   *(See Reviewer 3JTt W1&Q1, 5R5q W2)*

2. **Explaining Performance Differences Across Grounding Datasets**
   We clarify why TokenSculpt improves significantly on ActivityNet-Grounding but shows limited change on Charades-STA. The key difference lies in redundancy levels: Charades videos are short and low-redundancy (high compression sensitivity), while ActivityNet contains substantial redundancy (compression helps expose key frames).
   *(See Reviewer 3JTt W2&Q2, rAus W1)*

3. **Efficiency Analysis under Multiple Pruning Ratios**
   We add detailed runtime, memory, TFLOPs, and speedup comparisons under 33%, 49%, and 66% token retention across ToMe, DART, and TokenSculpt. Results demonstrate **consistent speedup trends**, confirming TokenSculpt’s stable efficiency across pruning strengths.
   *(See Reviewer 3JTt W3&Q3)*

4. **Theoretical and Empirical Evidence for Min–Max Anchors**
   We provide full theoretical justification for why Min–Max anchors encode global spatiotemporal boundaries, outperforming attention-based or similarity-based merging in preventing boundary loss. Grounding experiments and visualizations (e.g., panda boundary in Fig. 2) further validate that TokenSculpt preserves global structure.
   *(See Reviewer 3JTt W4&Q4, KDoL W3)*

5. **Long-Video Understanding and Extended Temporal Context Analysis**
   Experiments on VideoMME-Long, EgoSchema, and LongVideoBench (>10 minutes) show that TokenSculpt continues to improve performance even for extremely long videos (up to 1 hour), indicating **no inherent limitations in long-video reasoning**.
   *(See Reviewer 5R5q W1, KDoL W1)*

6. **Performance–Efficiency Trade-off and Performance/Time Ratio Analysis**
   Beyond simple accuracy or speed comparison, we introduced the performance/time ratio. TokenSculpt improves this ratio by **1–2%** overall and **5% on grounding tasks**, while ToMe’s minor advantage on QA benchmarks does not reflect true differences since those benchmarks lack spatial/temporal evaluation.
   *(See Reviewer 5R5q W3)*

7. **Runtime Overhead of Bipartite Matching in Uniform Sampling**
   We provided a practical runtime study of the bipartite matching step. The cost is 0.0004–0.001s per video on A800 GPUs—comparable to a single MatMul—thus negligible in practice. Implementation details are provided in the Supplement.
   *(See Reviewer KDoL W2)*

8. **Clarification of Positional Encoding Interaction**
   Since TokenSculpt explicitly manipulates token order, we clarify how positional encodings (e.g., RoPE in Qwen2.5-VL) are regenerated using updated (t, h, w) coordinates. Without reordering, grounding performance drops significantly. We also align ToMe implementations to ensure fair comparison.
   *(See Reviewer KDoL Q1)*

9. **Boundary Identification and Irregular Region Handling**
   We explain how Min–Max anchors remain valid even for irregular or non-rectangular regions by enclosing merged tokens within a tube and selecting extrema along temporal and spatial axes. Grounding results (+3.6% mIoU over ToMe at 33%) demonstrate accurate boundary preservation.
   *(See Reviewer KDoL Q2)*

10. **Extending TokenSculpt with Dynamic Anchor Allocation**
   We show that TokenSculpt can be naturally extended to dynamically adjust anchor numbers according to scene redundancy or complexity, supported by similarity-based redundancy factors defined along spatial and temporal axes.
   *(See Reviewer KDoL Q3)*

**Overall, we believe we have fully addressed all reviewers' concerns.**

Finally, we would like to thank the ACs for their efforts in handling this unexpected situation. We believe our contributions have great potential and can inspire further exploration in the field of token pruning.

Best regards,

Authors of Paper #24422

---

### Meta-Review · Area_Chair_cjoN · 2026-01-06

**Summary:**

The paper presents TokenScuplt, a training-free token pruning and merging method for multimodal large language models that aims to reduce inference cost while preserving the spatio-temporal structure of visual tokens. The paper received mixed initial reviews. The reviewers recognized the strong empirical results and their practical implications, yet they raised several concerns. Common issues included: (1) insufficient explanation of how the core min-max mechanism preserves spatio-temporal structure; (2) limited empirical validation and justification, including inconsistent results on Charades-STA and missing evaluations across different token retention rates and longer video benchmarks; and (3) unclear technical details in parts of the method. While some of these concerns were addressed in the rebuttal, the explanation of min-max mechanism remains largely unaddressed.

The AC also notes ongoing concerns regarding the clarity of the technical presentation. Specifically, Lines 187-191 describe TokenScuplt as consisting of three steps: (1) identifying anchor tokens via min-max duplication, (2) merging sub-sampled tokens, and (3) sorting the merged tokens. However, it is unclear how anchor tokens are selected in practice. Min-max duplication is defined over a tube formed by a set of tokens; if this set corresponds to the original input tokens, the resulting anchors appear to be limited to corner tokens of the input video or its crops. If additional inputs, such as key spatio-temporal boundaries, briefly mentioned in Line 230, are required, these are not clearly described in the main paper or appendix. In addition, bipartite matching is mentioned in the abstract and discussed in the appendix, but its role is not clearly explained in the main paper, making it unclear when and how it is applied within the proposed framework. While ACs are not supposed to provide independent reviews, it is their responsibility to ensure clarity and rigor in accepted papers.

Given the remaining concerns, particularly around conceptual explanation and technical clarity, the AC cannot recommend acceptance at this time, despite the current ratings. The authors are encouraged to further clarify the core mechanisms, strengthen the technical exposition, and consider resubmitting the work to a future venue.

**Reviewer Concerns:**

In their rebuttal, the authors provided additional clarifications of the technical components and conducted further experiments. These additions largely addressed the concerns regarding empirical performance and validation. However, the central concern regarding the min–max mechanism remains insufficiently explained. While the authors clarified the definition of min-max anchors and demonstrated empirical benefits, the rebuttal does not clearly articulate how these anchors preserve spatio-temporal structure. The only explanation provided in the rebuttal states that "(these anchors) explicitly preserves the extremal boundary tokens, preventing both bottom-bias and huge-cluster issues, and maintaining the global structure after pruning". This does not fully address the underlying issue.

**Reviewer Scores:**

Due to the remaining concern on the min-max mechanism, the AC expects that Reviewer 3JTt and KDoL, who shared this concern, may maintain or lower their ratings.

---

### Decision · Program_Chairs · 2026-01-26

Reject